# Does Acupuncture Hurt? A Retrospective Study on Pain and Satisfaction during Pediatric Acupuncture

**DOI:** 10.3390/children10111774

**Published:** 2023-11-01

**Authors:** Jeffrey I. Gold, Monika Kobylecka, Nhat H. Ngo, Christopher T. Lin, Caitlyn N. Hurray

**Affiliations:** 1Departments of Anesthesiology, Pediatrics, and Psychiatry & Behavioral Sciences, Keck School of Medicine, University of Southern California, Los Angeles, CA 90089, USA; 2Department of Anesthesiology Critical Care Medicine, The Saban Research Institute at Children’s Hospital Los Angeles, Los Angeles, CA 90027, USAhurray@usc.edu (C.N.H.); 3Keck School of Medicine, University of Southern California, Los Angeles, CA 90089, USA

**Keywords:** acupuncture, needling pain, pediatric pain, chronic pain, integrative medicine, patient satisfaction, anxiolysis, Kiiko Matsumoto Style (KMS)

## Abstract

Previous studies have shown the benefit and safety of pediatric acupuncture, but it is often rejected by patients and their caregivers due to the perception of needling pain associated with acupuncture. A retrospective cohort study of 230 unique patients (1380 sessions) aged 8 to 21 underwent Kiiko Matsumoto Style acupuncture in an outpatient pain clinic. Patients completed a post-acupuncture survey, including the Faces Pain Scale-Revised and Likert-like scales about overall satisfaction, relaxation, and anxiolysis. Univariate analyses were conducted on all outcomes of interest. The mean needling pain score was 1.3 out of 10 with 57.7% of patients reporting no needling pain during their first acupuncture session. The mean score for overall satisfaction was 8.4 out of 10, relaxation was 8.2 out of 10, and anxiety reduction was 7.7 out of 10. The overall satisfaction, relaxation, and anxiolytic effect of acupuncture was increased in patients with more sessions (*p* = 0.003, 0.022, 0.004, respectively). There was no change in needling pain scores in patients with an increased number of acupuncture sessions (*p* = 0.776). Patients experience minimal needling pain during acupuncture needling and are highly satisfied with acupuncture. Those with more treatment sessions report feeling increased satisfaction and relaxation.

## 1. Introduction

Integrative approaches to pediatric pain management have been gaining attention and popularity, as they have the potential to help children manage symptoms (i.e., pain, anxiety) associated with chronic pain, as well as decrease the usage of addictive opioid medications [1]. While acupuncture therapy has been routinely prescribed clinically with adults and less often with children, most research to date has focused on the safety and efficacy of acupuncture for patients with various pain conditions. In children, acupuncture has been shown to be effective and safe with a limited number of adverse events. These include minor events like transient bleeding and numbness to more severe events like infection and pneumothorax [2,3]. Multiple studies and systematic reviews have found that acupuncture has broad and promising benefits for various types of pain conditions, including acute and chronic pain, migraine headaches, sickle-cell disease, and procedural pain [4,5,6].

Despite evidence on the safety and efficacy of pediatric acupuncture for different pain pathologies, acupuncture is often rejected by pediatric patients and their caregivers due to the perception of needling pain associated with acupuncture [7]. Similarly, pediatricians do not often consider acupuncture for pediatric pain patients because they believe that children may be unwilling to undergo short-term discomfort for long-term pain relief [8]. While many studies explore pain levels in pediatric patients before and after acupuncture sessions, only one study has examined self-reported acupuncture–related pain and satisfaction during acupuncture needling [8].

The purpose of this study is to explore pediatric patients’ self-reported experiences with acupuncture with a primary focus on needling pain, anxiolysis, acupuncture satisfaction, and relaxation. A secondary aim of this study is to investigate whether the number of acupuncture sessions is related to any changes in patient-reported needling pain, anxiolysis, satisfaction, and relaxation. A large sample of outpatients were enrolled from a pain medicine clinic with varying pain conditions, severity, and symptomology. It was hypothesized that acupuncture would be a minimally painful, anxiety-reducing, satisfying, and relaxing experience. It was also hypothesized that the experience would be more favorable in patients who attended more acupuncture sessions. The main clinical implication of the current study is to demonstrate that acupuncture is minimally painful and that patients report positive outcomes associated with acupuncture. The hope is that the results can aid in educating healthcare providers, patients, and caregivers on the relatively pain-free nature of acupuncture and allow all parties to make an informed decision on the role of acupuncture as an acceptable integrative medicine practice for pediatric patients and part of a comprehensive treatment paradigm that focuses on whole person health.

## 2. Methods

### 2.1. Setting

The current retrospective cohort study focuses on an outpatient pain clinic convenience sample of pediatric patients receiving acupuncture for chronic pain complaints. Patients were assessed and referred for acupuncture after their intake visit by the pain clinic team. Routine surveys were collected after acupuncture sessions as part of the acupuncture clinic visit to assess acupuncture needling pain and overall satisfaction with the acupuncture experience. Patient medical records from the pain clinic were accessed to review the routine surveys and data collected under the standard of care procedure for non-research purposes. Patient data was collected and analyzed from the first 5 years (January 2009 to December 2013) of this inaugural acupuncture program. All the acupuncture sessions and data collection were conducted at an urban, pediatric, and academic medical center in the western US. This manuscript adheres to applicable STROBE guidelines. The study was reviewed and approved by the local hospital Institutional Review Board (IRB) committee.

### 2.2. Study Population

A total of 236 unique pediatric patients were identified with completed surveys. Data was reviewed from all patients who attended an acupuncture session at least once and completed the self-report acupuncture questionnaire as part of the standard of care procedure. Caregivers were also given the opportunity to leave a comment at the end of the questionnaire. Six patients who were younger than 8 years old at the time of survey completion only had caregiver-completed surveys and were excluded from the analysis. This study includes data from 230 unique pediatric patients (1380 total treatment sessions) from patients aged 8 to 21 (mean age = 15.9 years old). The study was set at a pediatric hospital that treats patients from 0 to 21, and as a result, our population will be referred to as pediatric. The number of acupuncture sessions attended was not controlled, with a range of 1 to 71 sessions. There was a total of 1380 completed questionnaires included in the analysis with a median of 3 acupuncture sessions per patient. This retrospective convenience cohort reflected the ethnic diversity of the medical center with 67% Latinx, 24% White, 4% Black, 3% Asian, and 2% Other. The patient population was comprised of a heterogeneous patient population with a combination of primary/functional pain disorders (i.e., fibromyalgia, complex regional pain syndrome) and comorbid musculoskeletal chronic pain conditions to another primary (i.e., sickle cell disease, cancer, and juvenile idiopathic arthritis) medical condition. All patients were English-speaking.

### 2.3. Acupuncture Procedure

The acupuncturists, employed at the pain clinic, used Kiiko Matsumoto Style (KMS), a type of Japanese acupuncture to treat all patients who agreed to receive treatments [9,10]. Each treatment was individualized to the patient using KMS protocol of identifying active reflexes through systematic palpation of the abdomen, neck, channels, and specific points on the feet. This diagnostic evaluation allowed for the identification of patterns associated with the patient’s presenting symptoms and facilitated the selection of treatment points. Changes in both the active reflexes and the patient’s presenting symptoms indicate a successful treatment.

The needles used were sterile, single-use, disposable, Seirin (brand) 40-gauge (0.16 mm wide) needles inserted at depths from 5 to 15 mm depending on the location of the point and the size of the patient. The total number of needles used was dependent on the number of identified positive, KMS reflexes. The needle retention time, which increased with age, was at least one minute but did not exceed twenty minutes.

### 2.4. Measures

After each acupuncture session, patients completed an acupuncture questionnaire as part of the standard of care procedure. To avoid observer bias, the questionnaire was completed in the front office of the clinic during checkout, away from the acupuncturist. The acupuncture questionnaire is a 6-item Likert-like measure used to assess overall satisfaction, relaxation, anxiety reduction, and satisfaction with the acupuncturist. The questionnaire also includes an acupuncture pain score measured with the Faces Pain Scale-Revised, which is a validated measure of pain in children [11]. The questionnaire [Appendix A] was completed on paper and responses were de-identified and entered into IBM SPSS Statistics 28.0 software for analysis.

### 2.5. Statistical Analysis

Power analyses based on preliminary data from the principal investigator’s previous acupuncture studies determined that 200 patients were needed for this study. Baseline pain characteristics were not collected as part of the standard of care procedure and were not available to analyze.

Primary outcomes include pain associated with needling, anxiety reduction, pain reduction, relaxation, and satisfaction scores. The variables were analyzed both individually and across treatment cohorts. For the grouped data analysis, patients were split into 4 groups; A: patients who had only gone to 1 acupuncture session, B: patients who had gone to 2 or 3 sessions, C: patients who had gone to 4 to 6 sessions, D: patients who had gone to 7 or more sessions. Treatment group criteria/assignments were delineated based on the acupuncturist’s recommendation on the number of treatments that would yield incremental increases in treatment response.

Univariate analyses of primary outcomes were conducted through IBM SPSS Statistics 28.0. The distribution (median, quartiles, mean, standard deviation) of needling pain, anxiety reduction, relaxation, and satisfaction scores across all patients in their first acupuncture session and final acupuncture session were calculated. Due to the skewed distribution of the primary outcomes, nonparametric statistical analyses were applied. Kruskal-Wallis and Dunn’s post-hoc test with Bonferroni correction analyses were conducted to analyze intergroup comparisons of needling pain, anxiety reduction, relaxation, and satisfaction scores across each cohort. Wilcoxon Signed Rank tests were conducted to analyze inter-session differences between the first and final sessions, if applicable. All *p*-values were assessed at the 0.05 significance level.

## 3. Results

A total of 230 unique patients (76% female) and 1380 completed acupuncture questionnaires were analyzed with an overall mean age of 15.9 years old [Table 1]. Results indicate that on average patients experienced minimal needling pain during the acupuncture session (M = 1.3, SD = 2.0) on the Faces Pain Scale-Revised. Fifty-eight percent (128/222) of patients reported that they experienced no pain associated with the acupuncture needling practice [Figure 1]. Twenty-eight percent (63/222) of patients reported that they experienced mild (2 out of 10) pain associated with acupuncture needling. Patients had positive experiences during their first acupuncture session [Table 2] reporting that acupuncture was good (M = 8.4, SD = 2.2), relaxing (M = 8.2, SD = 2.3), and anxiety-relieving (M = 7.7, SD = 2.6). Moreover, 153 out of 223 (68.6%) patients felt that acupuncture helped decrease their existing pain, while 63 out of 223 (28.3%) were unsure and 7 out of 226 (3.1%) reported no pain reduction. Patients were highly satisfied with the acupuncturist, with a mean score of 9.1 out of 10 (SD = 1.5).

The sex of the patient had no impact on overall satisfaction (*p* = 0.777), relaxation (*p* = 0.096), anxiety reduction (*p* = 0.733), needling pain (*p* = 0.642), and acupuncturist satisfaction (*p* = 0.220). The presenting pain type (musculoskeletal, headache, abdominal, generalized/other) did not influence the mean first session acupuncture score for overall satisfaction (*p* = 0.096), relaxation (*p* = 0.260), anxiety reduction (*p* = 0.671), needling pain (*p* = 0.712), and acupuncturist satisfaction (*p* = 0.902). Further, there were no differences in sex (*p* = 0.491), gender (*p* = 0.682), and presenting pain type (*p* = 0.804) between cohorts.

Patients reported high scores during their final acupuncture treatment session for overall satisfaction (M = 8.5, SD = 2.2), relaxation (M = 8.4, SD = 2.3), anxiety reduction (M = 7.8, SD = 2.9), and satisfaction with the acupuncturist (M = 9.0, SD = 1.9) and low scores on pain associated with acupuncture needling (M = 1.4, SD = 2.1). While the overall satisfaction with acupuncture and its anxiolytic effect has increased mean scores when compared to the initial acupuncture session, needling pain and acupuncturist satisfaction remain positive and relatively unchanged. When separated by cohort (A through D), there appears to be a trend of higher scores in the most recent acupuncture session when patients attended more acupuncture sessions, as seen in Table 3. The *p*-value for a Kruskal-Wallis test indicates significant differences between acupuncture session frequency and overall satisfaction (*p* = 0.034), relaxation (*p* = 0.042), and anxiolysis (*p* = 0.009) [Table 3]. No significant difference was seen in cohorts with increased acupuncture session frequency for needling pain (*p* = 0.776) and acupuncturist satisfaction (*p* = 0.832) [Table 3]. A post-hoc Dunn- Bonferroni analysis of the data reveals that the largest difference occurs between patients who have had single sessions (cohort A) and those who have had four or more sessions (cohorts C and D), which can be seen in Appendix B. Additionally, Patients reported significantly higher relaxation (*p* = 0.035) and anxiolysis (*p* = 0.030) scores on average compared to their initial session [Table 4].

One patient had one adverse effect during one acupuncture session. The patient’s mild facial redness and swelling resolved on its own with no impact on breathing or eating at the conclusion of the session. No other adverse events were reported or observed.

## 4. Discussion

The current findings support the hypothesis that KMS acupuncture is relatively pain-free and well-liked in a pediatric patient population. Acupuncture-related pain from needling remains low to none irrespective of whether it is an initial acupuncture treatment session or a follow-up treatment session. While pain and fear of acupuncture is a commonly held perceptions among patients, caregivers, and healthcare providers, the current study shows that acupuncture is relatively pain-free. The majority (87%) of patients report experiencing minimal to no needling pain during acupuncture. Given the current findings paired with the literature on the benefits of acupuncture, healthcare providers can feel more confident that KMS acupuncture does not hurt, is well received, and has evidence supporting a variety of positive health outcomes. Healthcare providers can also educate and dispel the commonly held perception that acupuncture is painful. Additionally, patients and their caregivers reported high overall satisfaction, relaxation, and reductions in anxiety. Moreover, most patients report a reduction of their self-reported pain symptoms immediately following acupuncture, regardless of their presenting pain problem.

The results indicate a significant change in primary outcomes, reflecting higher levels of overall satisfaction, relaxation, and anxiety reduction in cohorts that participate in more acupuncture sessions. Although patients experienced significantly greater relaxation and anxiety reduction between their initial and final acupuncture sessions [Table 4], the current study demonstrated that optimal outcomes were achieved for patients following their fourth and sixth acupuncture treatment sessions [Appendix B]. However, overall treatment session cohorts needling pain remained consistently low (median = 0), and acupuncturist satisfaction scores were reliably high (median = 10).

The current findings contribute to the growing evidence on pediatric acupuncture reflecting zero to minimal self-reported pain associated with needling, acceptability and satisfaction, and the effectiveness of acupuncture in patients with chronic pain. While most studies examine pain ratings before and after acupuncture sessions related to the patient’s presenting health condition, the current study uniquely and primarily examines the self-reported pain experience of the acupuncture needling itself, while also investigating satisfaction and other therapeutic benefits. Kemper and colleagues in a smaller cohort previously found that acupuncture was helpful and acceptable in patients with severe chronic pain as reported by the patient’s guardians over a telephone interview [8]. The current study extends these findings with a larger cohort, reflecting a heterogenous pain sample, and the use of psychometrically sound self- and caregiver-reported outcomes, thus increasing the generalizability of acupuncture for pediatric chronic pain management. Additionally, many of the patients received more than one acupuncture session, allowing for analyses to be conducted on four cohorts of varying numbers of treatment sessions. Emerging research in this area indicates a strong potential for acupuncture and other integrative medicine practices to be used for pain management in children.

The study also gathers favorable qualitative comments from patients and their caregivers, such as a 16-year-old patient stating, “I was scared at first, but I found it to be an enjoyable experience. I was very relaxed as the result with it (acupuncture)”. Another 16-year-old patient reports, “It always helps me to feel better and relaxed. I never feel any pain or discomfort either”. A mother of a 15-year-old patient claims, “Acupuncture has helped my son tremendously. No pain meds are needed at all”. Overall, the acupuncture service is received with overwhelmingly positive feedback and satisfaction.

The present study has some limitations. First, the study only examines patient data from children aged 8 or older because that is the age at which they could self-report on the acupuncture questionnaire. Thus, results cannot be applied to children younger than 8 years. Second, patients are more likely to attend more acupuncture sessions if they are satisfied with the initial sessions, which may influence the ratings of sequential acupuncture sessions. Further, our study is subject to self-selection bias as participants who were unsatisfied with acupuncture are less likely to continue with treatment. Lastly, Kiiko Matsumoto Style acupuncture, a known gentler acupuncture technique, was used for all patients, therefore the findings may not generalize to all forms of acupuncture, like other Traditional Chinese Medicine acupuncture practices.

Future investigations examining the effects of acupuncture for chronic pain in children would benefit from randomized-controlled clinical trials with younger cohorts, comparisons to other integrative medicine practices, like massage or yoga, and pharmacological treatment options. Integrative care is what is required to manage chronic pain and the associated health and co-morbid mental health issues. Ultimately conducting clinical trials that take a more pragmatic control trial (PCT) approach, may have greater generalizability to real-life struggles and solutions for patients with complex pain and associated medical symptoms. In the end, patients require multi-modal therapies to tackle the complexities associated with chronic pain, therefore engaging in PCT may ultimately shed light on the necessary treatment combinations for doing exactly what we’ve named our pain management clinics, pain “management”.

Overall, acupuncture is a promising and relatively pain-free strategy that can be used to treat patients with chronic pain, focus on pain and anxiety reduction, and help patients feel more relaxed and satisfied with their treatment. Mixed method data collection, reflecting quantitative and qualitative data, strongly suggests that patients are highly satisfied with their acupuncture treatment experience. The data dispels the belief that acupuncture needling is painful and should signal physicians, caregivers, and patients to recommend/try acupuncture for the management of pediatric pain symptoms and other pain-associated concerns. The mandate to hoist integrative medicine practices to the highest level for patients suffering from chronic pain has never been greater, given the complexities of their medical problems and the co-morbidity of mental health concerns. It is upon all healthcare providers to recognize the debilitating nature of chronic pain, embrace a multi-modal integrative treatment approach, and confidently recommend acupuncture in an effort to reduce patient suffering, increase quality of life, and improve other pediatric health outcomes.

## Figures and Tables

**Figure 1 children-10-01774-f001:**
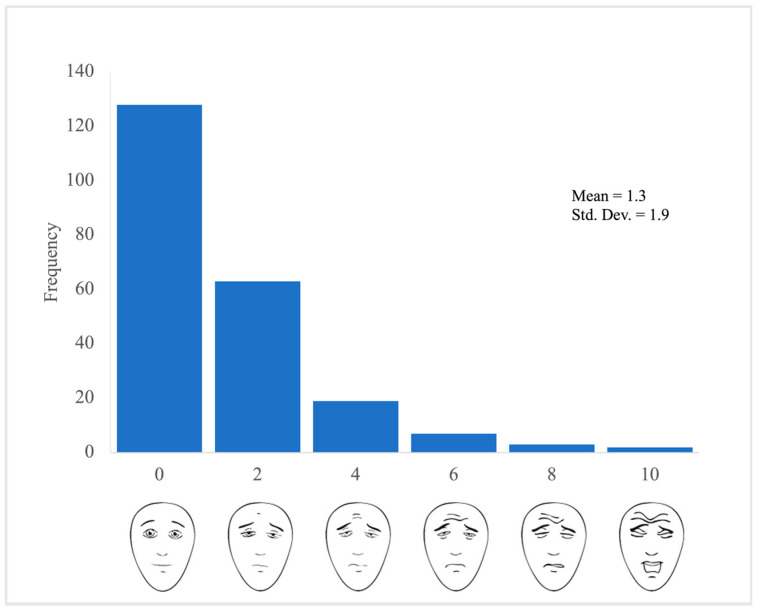
Faces Pain Scale-Revised Frequency Plot for First Acupuncture Session (*n* = 222). This frequency plot includes data from patients’ first acupuncture sessions at the pain clinic. All responses were self-reported by the patients.

**Table 1 children-10-01774-t001:** Patient Demographic Data and Chief Complaint.

**Mean Age**	15. 9 (SD = 2.8)
**Median Number of Sessions**	3 (25th percentile: 1, 75th percentile: 6)
*n* = 230	n (%)
**Sex**	
Females	175 (76.1%)
**Pain Type**	
Musculoskeletal Pain	112 (48.7%)
Headache	14 (6.1%)
Abdominal Pain/Nausea	11 (4.8%)
Generalized/Multiple Pain Sites	93 (40.4%)
**Cohort by Number of Acupuncture Sessions**	
1 Session (A)	66 (28.7%)
2–3 Sessions (B)	69 (30.0%)
4–6 Sessions (C)	42 (18.3%)
7+ Sessions (D)	53 (23.0%)

**Table 2 children-10-01774-t002:** Patient First Session Data.

	Median	1st Quartile	3rd Quartile	Mean	Standard Deviation
Overall Satisfaction	9	7	10	8.4	2.2
Relaxation	9	7	10	8.2	2.3
Anxiety Reduction	9	6	10	7.7	2.6
Pain	0	0	2	1.3	2.0
Acupuncturist Satisfaction	10	9	10	9.1	1.5

**Table 3 children-10-01774-t003:** Kruskal Wallis comparison of primary outcomes of patients’ most recent acupuncture sessions between session cohorts where “A” indicates 1 session, “B” as 2 to 3 sessions, “C” as 4 to 6 sessions, and “D” as 7 or more sessions.

Measure	Session Cohort	*n*	Mean	SD	Mean Rank	χ^2^	*p*-Value
Overall	A	67	8.04	2.44	101.78	8.66	0.034 *
Satisfaction	B	68	8.22	2.37	108.97		
	C	42	9.04	1.40	127.96		
	D	53	9.37	1.47	131.21		
Relaxation	A	64	7.94	2.47	96.95	8.19	0.042 *
	B	66	8.29	2.42	111.62		
	C	42	8.91	1.86	129.67		
	D	53	9.10	1.58	120.61		
Anxiety	A	65	7.08	3.02	94.02	11.63	0.009 *
Reduction	B	63	7.52	3.03	105.35		
	C	41	8.71	1.90	129.99		
	D	52	8.73	2.38	123.88		
Pain	A	66	1.33	2.09	108.57	1.11	0.776
	B	66	1.55	2.35	115.20		
	C	41	1.41	1.75	117.44		
	D	50	1.24	2.02	107.71		
Acupuncturist	A	67	8.73	2.15	111.12	0.87	0.832
Satisfaction	B	67	8.88	1.94	112.29		
	C	42	9.07	1.91	121.37		
	D	52	9.43	1.19	116.13		

* *p* < 0.05. See Appendix B for pairwise comparisons.

**Table 4 children-10-01774-t004:** Wilcoxon signed a rank test between the first and final acupuncture sessions of those who had more than one acupuncture session.

Measure ^1^	Ranks ^2^	N	Mean Rank	Sum of Ranks	Z	Asymp. Sig.
Overall	Neg.	38	47.30	1797.50	−1.798	0.072
Satisfaction	Pos.	57	48.46	2762.50		
	Ties	68				
Relaxation	Neg.	43	46.41	1995.50	−2.113	0.035 *
	Pos.	59	55.21	3257.50		
	Ties	56				
Anxiety	Neg.	32	53.02	1696.50	−2.171	0.030 *
Reduction	Pos.	63	45.45	2863.50		
	Ties	60				
Pain	Neg.	29	32.33	937.50	−0.510	0.610
	Pos.	34	31.72	1078.50		
	Ties	89				
Acupuncturist	Neg.	44	40.10	1764.50	−0.492	0.623
Satisfaction	Pos.	37	42.07	1556.50		
	Ties	80				

^1^ Most recent session minus initial session score. ^2^ Negative, positive, or tie indicates the direction of the difference. * *p* < 0.05.

## Data Availability

The data presented in this study are available on request from the corresponding author. The data are not publicly available due to patient privacy.

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
