# Peer review of "Does Acupuncture Hurt? A Retrospective Study on Pain and Satisfaction during Pediatric Acupuncture"

_children, 2023, doi:10.3390/children10111774_

Round 1

Reviewer 1 Report

Comments and Suggestions for Authors

I commend you for studying a topic that desperately needed studying. This is certainly very important to the literature. While I acknowledge you have mentioned that by only using KMS acupuncture in your study, that these results may not be generalizable to TCM, I actually believe that they cannot be generalizable to the much more common TCM-style acupuncture. There is too much difference between traditional acupuncture and KMS. Therefore, I disagree with your statement that "... current findings support the hypothesis that acupuncture is relatively pain-free and well-liked in a pediatric patient population." I believe it is only accurate to say that the current findings support the hypothesis that KMS acupuncture is relatively pain-free and well-liked in a pediatric patient population".

I also have a question about overlapping pain. For example, nobody had both MSK pain AND headache? Or were having two conditions considered under the "generalized / multiple pain sites" category? This wasn't clear to me. Similarly, did MSK pain only mean MSK pain in a single muscle or body part? 

Comments on the Quality of English Language

line 90 - did you mean to say "convenience" cohort instead?

line 93 - "reginal" should be "regional"

line 131 - add "on" after "based"

line 213 - what do you mean by "over treatment sessions"?

line 254 - PCT generally refers to "pragmatic clinical trial" but "pragmatic controlled trial" is also acceptable, rather than "practical control trial"

Author Response

Dear reviewer,

Thank you for your praise and critiques of our paper. We have implemented the following changes to our paper based on your recommendations:

  • Clarified in our discussion section that our data supports “KMS acupuncture”
  • Made edits to spelling mistakes listed in the review
  • Added clarification to line 330, “over all treatment session cohorts…”
  • Changed PCT to pragmatic control trial

To answer your question about MSK and headache, if patients had reported two or more pain locations/sites, they would be considered under the “generalized / multiple pain sites” category. If you believe we should add a footnote to Table 1 to clarify this point, please let us know.

In addition to your comments, we would like to inform you that we have re-evaluated our statistical analyses based on another reviewer’s comments. Due to the skew of our data and the violation of equally distributed data, we have employed nonparametric statistical approaches. We re-analyzed our primary findings using Kruskal-Wallis tests (instead of ANOVA) and Wilcoxon Signed Rank test (instead of paired t-tests). Thus, we have also updated our section 2.5. Statistical Analysis. Our new findings replace Table 3, Table 4, and Appendix B. These new analyses maintained most of the original findings and yielded a few additional significant findings. Of note, additional findings indicated significance within groups comparing first and final acupuncture sessions in patients with regard to relaxation and anxiolysis. We will upload a clean copy of the new manuscript, but please feel free to request a copy with tracked changes.

Thank you for your continued collaboration,

Study Team

Reviewer 2 Report

Comments and Suggestions for Authors

As authors opinion, acupuncture is a promising and relatively pain-free strategy that can be used to treat pediatric patients with chronic pain to reduce pain and anxiety, and help patients feel more relaxed and satisfied with their treatment, For multi-modal integrative treatment approach, to reduce patient suffering and increase quality of life and other positive pediatric health outcomes. acupuncture is highly and confidently recommended. It’s strongly agreed points. To recommend and apply acupuncture for treatment in many more areas than now, the following concerns and suggestions have to be considered more in detail, and it needs to be revised for the quality of manuscript with clear clarification.

1) The study was conducted in accordance with and approved by the Institutional Review Board of Children’s Hospital Los Angeles (protocol code CCI-13-00115 approved April 5th, 2013). and . Patient data was collected and analyzed from the first 5 years (January 2009 to 74 December 2013) of this inaugural acupuncture program : Now, it’s 2023, it’s like this study was done 20 years ago. All the references are before 2015 except reference [1]. it has to be clarified the reason to be used the data and IRB approved research 20 years ago. it would be fine to use the data 20 yeas ago, but IRB approval date looks too old. The current trend has to be considered more in detail.

2) In this paper, the age is described as 9 to 21 years old, and I would like to ask if the age corresponds to the age of pediatric patient inclusion..

3) it is important to determine the severity of the pain when a pediatric patient first comes to the outpatient clinic due to pain. There’s no description in the manuscripts. As concerned knowledge, .If the pain is higher than the needling pain, the needling pain may not be felt. It’s strongly recommended to describe the severity of pain when cames first.

4) In Statistical analysis and Results, the followings are recommended to revise and improve the manuscript ;

SPSS version and Exact name of software has to be described instead of only describing SPSS.

Authors described in 2.3 .Acupuncture Procedure - ‘needles inserted at depths from 5 to 15 mm, needle retention time was at least one minute but did not exceed twenty minutes’- In table 2.1, it’s suggested that depths and retention time has to be inclued for the acupuncture  treatments.  As importantly, the 1st pain severity has to be included in table 2.1, Moreover, to compare measures among 3 groups(B, C, D in table 2), it has to be clarified no difference for the measures( age, sex, pain type, 1st pain severity, needle depth, retention time).   

In table3, it’s the results of ANOVA among B, C, D( it’s grouped by the The numbers of session). For this, it’s strongly recommended to analysze ANCOVA with covariates instead of only ANOVA. If it’s clarified no difference for the measures for the measures( age, sex, pain type, 1st pain severity, needle depth, retention time), it’s available to use the results of ANOVA.

 For the description ‘A linear regression analysis was conducted to gauge the correlation between needling pain scores and satisfaction scores, resulting in an R2 of 0.104.’ ,  

it’s not sure why linear regression analysis was used for the correlation between needling pain scores and satisfaction scores. Only correlation coefficient would be enough for it. If authors want to know R2(here, it seems 0.104 and too low explanation), the results of linear regression analysis has to be described even in supplementary materials.

For table 4 ; 

With Table 1, 2 and 3, It’s not clear that there’s no difference between first and final acupuncture session for the measures. Evenif there’s no difference by paired t-test for all patients with all measures, it’s highly recommened to do repeated measures two factor analysis for B, C ,D groups with all measures(Overall satisfaction, relaxation, Anxiety reduction, pain, Acupuncturist satisfaction).

 5) In Discussion, 

Authors described ‘The results -----no correlations between needling pain and satisfaction during acupuncture sessions.’ - the result is not surely presented in the manuscript. it would be better to be presented in manuscript if the sentence is included in discussion.   

6) In manuscript ;

- p score has to be changed as p-value

-Some sentences are not clear. As am example, in page 8, ‘the patient’s caregivers over a telephone interview.8’ -is it correct sentence ?

-Others has to be checked in detail and revise the sentences if need.

Thank you so much.

Comments on the Quality of English Language

As authors opinion, acupuncture is a promising and relatively pain-free strategy that can be used to treat pediatric patients with chronic pain to reduce pain and anxiety, and help patients feel more relaxed and satisfied with their treatment, For multi-modal integrative treatment approach, to reduce patient suffering and increase quality of life and other positive pediatric health outcomes. acupuncture is highly and confidently recommended. It’s strongly agreed points. To recommend and apply acupuncture for treatment in many more areas than now, the following concerns and suggestions have to be considered more in detail, and it needs to be revised for the quality of manuscript with clear clarification.

1) The study was conducted in accordance with and approved by the Institutional Review Board of Children’s Hospital Los Angeles (protocol code CCI-13-00115 approved April 5th, 2013). and . Patient data was collected and analyzed from the first 5 years (January 2009 to 74 December 2013) of this inaugural acupuncture program : Now, it’s 2023, it’s like this study was done 20 years ago. All the references are before 2015 except reference [1]. it has to be clarified the reason to be used the data and IRB approved research 20 years ago. it would be fine to use the data 20 yeas ago, but IRB approval date looks too old. The current trend has to be considered more in detail.

2) In this paper, the age is described as 9 to 21 years old, and I would like to ask if the age corresponds to the age of pediatric patient inclusion..

3) it is important to determine the severity of the pain when a pediatric patient first comes to the outpatient clinic due to pain. There’s no description in the manuscripts. As concerned knowledge, .If the pain is higher than the needling pain, the needling pain may not be felt. It’s strongly recommended to describe the severity of pain when cames first.

4) In Statistical analysis and Results, the followings are recommended to revise and improve the manuscript ;

➀ SPSS version and Exact name of software has to be described instead of only describing SPSS.

➁ Authors described in 2.3 .Acupuncture Procedure - ‘needles inserted at depths from 5 to 15 mm, needle retention time was at least one minute but did not exceed twenty minutes’- In table 2.1, it’s suggested that depths and retention time has to be inclued for the acupuncture  treatments.  As importantly, the 1st pain severity has to be included in table 2.1, Moreover, to compare measures among 3 groups(B, C, D in table 2), it has to be clarified no difference for the measures( age, sex, pain type, 1st pain severity, needle depth, retention time).   

➂ In table3, it’s the results of ANOVA among B, C, D( it’s grouped by the The numbers of session). For this, it’s strongly recommended to analysze ANCOVA with covariates instead of only ANOVA. If it’s clarified no difference for the measures for the measures( age, sex, pain type, 1st pain severity, needle depth, retention time), it’s available to use the results of ANOVA.

 ➃ For the description ‘A linear regression analysis was conducted to gauge the correlation between needling pain scores and satisfaction scores, resulting in an R2 of 0.104.’ ,  

➜ it’s not sure why linear regression analysis was used for the correlation between needling pain scores and satisfaction scores. Only correlation coefficient would be enough for it. If authors want to know R2(here, it seems 0.104 and too low explanation), the results of linear regression analysis has to be described even in supplementary materials.

➄ For table 4 ; 

With Table 1, 2 and 3, It’s not clear that there’s no difference between first and final acupuncture session for the measures. Evenif there’s no difference by paired t-test for all patients with all measures, it’s highly recommened to do repeated measures two factor analysis for B, C ,D groups with all measures(Overall satisfaction, relaxation, Anxiety reduction, pain, Acupuncturist satisfaction).

 5) In Discussion, 

Authors described ‘The results -----no correlations between needling pain and satisfaction during acupuncture sessions.’ - the result is not surely presented in the manuscript. it would be better to be presented in manuscript if the sentence is included in discussion.   

6) In manuscript ;

- p score has to be changed as p-value

-Some sentences are not clear. As am example, in page 8, ‘the patient’s caregivers over a telephone interview.8’ -is it correct sentence ?

-Others has to be checked in detail and revise the sentences if need.

Thank you so much.

Author Response

Dear reviewer,

Thank you for your helpful critiques of our paper. The following are our responses and implementations:

  1. The data was last collected ten years ago. We chose to do a literature search on pediatric needling pain from 2000 to current 2023 to determine if any studies had been done between 2000 to 2023 and the references reflect any acupuncture needle pain related studies in that period. There are three references in our manuscript that were published after 2015: 1, 5, and 7. The secondary analysis occurred on an IRB approved “data analysis only” protocol. The original protocol was submitted and accepted in 2013. According to IRB regulatory policies, data analysis only is permissible.
  2. This study was conducted in a pediatric hospital and the language frequently used for patients between the ages of 0 to 21 within a pediatric hospital is pediatric healthcare. However, the authors recognize that our cohort includes school aged, adolescents, and young adults. We have updated our language in 2.2. to clarify the broad usage of the term “pediatric.”
  3. Thank you for your great point on ways we could improve the protocol. Unfortunately, the current study did not focus on each patient's exact pain condition, but rather the survey asked about the experience/perception of pain associated with the acupuncture needle “poke” and the associated pain. Chronic pain is often associated with the perceived amplification of pain; therefore, this population would potentially experience needling pain at a higher level and not necessarily a lesser level. The data reflects that the needle associated pain was very low irrespective of the patients’ ongoing pain concerns. In addition, due to the retrospective nature of this study, we cannot retroactively extrapolate pain severity from this population.
  4. We greatly appreciate your assistance in critiquing our statistical analysis plan.
    • We have updated section 4. Statistical Analysis to clarify the version of SPSS we used.
    • The procedure outlined in 2.3. Acupuncture Procedure was for educational purposes for audiences to understand what the procedure consists of. As a pragmatic control trial, needle depth and time needled was determined solely by the acupuncturist and not by the research team. Similarly to point 3, due to the retrospective nature of the study, the exact depth and times for needling were not charted and/or controlled and cannot be added to our table. We did implement statistical analysis and did not find differences in age, sex, and pain type between cohorts and added that information to the results.
    • If we are understanding your question correctly, you believe that 1st pain severity, age, sex, pain type, depth, and retention should be considered are a covariate in the analysis. However, the current study is focused on needling pain and it is not a clinical trial determining the efficacy based on acupuncture procedure. Unfortunately, as a secondary analysis, we do not have the information of the depth, retention, and initial pain severity and cannot add these to the table. However, we have looked at differences by age, sex, and pain type and found that it is not significantly different between groups and within group.
    • This is a great point. We reviewed the manuscript and decided that the linear regression was ultimately not relevant to our manuscript, and we have removed it from our statistical analysis section as well as the associated results.
    • Thank you for your suggestion to use repeated measures two factor analysis on measures: overall satisfaction, relaxation, anxiety reduction, pain, and acupuncturist satisfaction. However, we are unclear regarding these suggestions. May you clarify your suggested analysis?
  5. Thank you for this much needed correction. Upon review, we recognize that the word “correlation” was misused as we were referring to the ANOVA results. To correct this, we replaced the world with “difference” as this is a more accurate depiction of an ANOVA test.
  6. All p-scores have been changed to p-value.

In addition to your comments, we would like to inform you that we have re-evaluated our statistical analyses. Due to the skew of our data and the violation of equally distributed data, we have employed nonparametric statistical approaches. We re-analyzed our primary findings using Kruskal-Wallis tests (instead of ANOVA) and Wilcoxon Signed Rank test (instead of paired t-tests). Thus, we have also updated our section 2.5. Statistical Analysis. Our new findings replace Table 3, Table 4, and Appendix B. These new analyses maintained most of the original findings and yielded a few additional significant findings. Of note, additional findings indicated significance within groups comparing first and final acupuncture sessions in patients with regard to relaxation and anxiolysis. We will upload a clean copy of the new manuscript, but please feel free to request a copy with tracked changes.

Thank you for your expertise and collaboration in this matter,

Study Team

Round 2

Reviewer 2 Report

Comments and Suggestions for Authors Thank you so much for the consideration on the suggestions and comments. 
However, it's required authors' opinion for the concerns below; 

1) In Table 3. authors described that “A” indicates 1 session, “B” as 2 to 3 sessions, “C” as 4 to 6 sessions, and “D” as 7 or more sessions.
If A, B, C, and D mean the repeated sessions for the same patients, it is suggested to do the ‘repeated measures factor analysis’.
It has to be clearly stated not to make confused in the manuscript

2) Table 4. Paired Wilcoxon signed rank test  ➜ it's better to delete 'Paired'

Author Response

Dear reviewer,

Thank you for your expertise and continued collaboration. We have made the following changes to our manuscript based on your comments

  1. Renamed Table 3 to reflect that the test was not a repeated measures factor analysis, but a between group comparison. More information on how we divided cohorts is listed in 2.5. Statistical analysis. Please let us know if these clarifications will suffice.
  2. We have renamed Table 4 and deleted ‘paired’.

Warm regards,

Study Team